# A systematic review of barriers and facilitators to antenatal screening for HIV, syphilis or hepatitis B in Asia: Perspectives of pregnant women, their relatives and health care providers

**Lucie Sabin** [ID]*, **Hassan Haghparast-Bidgoli**[☯], **Faith Miller**[ID][☯], **Naomi Saville**[ID][☯]

Institute for Global Health, University College London, London, United Kingdom

☯ These authors contributed equally to this work.
* lucie.sabin.21@ucl.ac.uk

**Data Availability Statement:** All relevant data are within the manuscript and its Supporting Information files.

## Abstract

### Background

Despite improvements, the prevalence of HIV, syphilis, and hepatitis B remains high in Asia. These sexually transmitted infections (STIs) can be transmitted from infected mothers to their children. Antenatal screening and treatment are effective interventions to prevent mother-to-child transmission (MTCT), but coverage of antenatal screening remains low. Understanding factors influencing antenatal screening is essential to increase its uptake and design effective interventions. This systematic literature review aims to investigate barriers and facilitators to antenatal screening for HIV, syphilis, and hepatitis B in Asia.

### Methods

We conducted a systematic review by searching Ovid (MEDLINE, Embase, PsycINFO), Scopus, Global Index Medicus and Web of Science for published articles between January 2000 and June 2023, and screening abstracts and full articles. Eligible studies include peer-reviewed journal articles of quantitative, qualitative and mixed-method studies that explored factors influencing the use of antenatal screening for HIV, syphilis or hepatitis B in Asia. We extracted key information including study characteristics, sample, aim, identified barriers and facilitators to screening. We conducted a narrative synthesis to summarise the findings and presented barriers and facilitators following Andersen's conceptual model.

### Results

The literature search revealed 23 articles suitable for inclusion, 19 used quantitative methods, 3 qualitative and one mixed method. We found only three studies on syphilis screening and one on hepatitis B. The analysis demonstrates that antenatal screening for HIV in Asia is influenced by many barriers and facilitators including (1) predisposing characteristics of pregnant women (age, education level, knowledge) (2) enabling factors (wealth, place of

**Funding:** The authors received no specific funding for this work.

**Competing interests:** The authors have declared that no competing interests exist.

residence, husband support, health facilities characteristics, health workers support and training) (3) need factors of pregnant women (risk perception, perceived benefits of screening).

## Conclusion

Knowledge of identified barriers to antenatal screening may support implementation of appropriate interventions to prevent MTCT and help countries achieve Sustainable Development Goals' targets for HIV and STIs.

## Introduction

Human immunodeficiency virus (HIV), syphilis and hepatitis B are sexually transmitted infections (STIs) that, if left undiagnosed and untreated, can lead to serious complications and death. Despite improvements in the last decade, their prevalence remains high in Asia [1, 2]. In 2017, 5.2 million people were living with HIV in the Asia Pacific region [3] and 123,000 people died from HIV-related causes in 2021 [4]. The regional prevalence of HIV was 0.2% [4]. In 2012, an estimated 1.8 million women were infected with syphilis in the South-East Asia region [5] and 39 million people with hepatitis B with a prevalence of 2.0% [6].

These STIs can be transmitted from infected mothers to their children during pregnancy and childbirth, resulting in significant morbidity and mortality. The rate of mother-to-child transmission of HIV in Asia and the Pacific is relatively high, at 17%, among the estimated 61,000 women living with HIV who gave birth in the region in 2017 [3] and 1.3 million pregnant women are at risk of transmitting HBV to their newborns each year [7]. The global number of adverse pregnancy events attributable to maternal syphilis infection was estimated to be 52,307 in the South-East Asia Region and 13,472 in the Western Pacific Region [8].

Mother-to-child transmission (MTCT), also called vertical transmission, can be prevented with simple and effective interventions, including antenatal screening and treatment, prevention of male-to-female transmission during sexual intercourse, and improving community awareness. Antenatal screening is an essential tool to enable women to find out if they are infected and to take the necessary steps to access preventive treatment if they test positive in order to avoid MTCT [9]. Since 2010, an estimated 7,400 new HIV infections among children in the Asia Pacific region were averted because of interventions aimed at reducing the MTCT of HIV [3]. However, due to limited availability and access to these interventions [10], antenatal screening for STIs in Asia remains low [11]. Only three of the 17 reporting countries in the Asia-Pacific region met the global target of over 95% coverage for knowledge of HIV status among women receiving ANC in 2017 and six countries (Bangladesh, Timor-Leste, Papuz New Guinea, Lao People's Democratic Republic, Indonesia, Singapore) reported coverage below 40% [11]. Only thirteen countries currently out of 17 countries have a policy of screening for hepatitis B during pregnancy, and very little data on hepatitis B screening coverage is currently available [10]. Most Asian countries also have no data on syphilis screening for pregnant women. Of the 28 countries in Asia and Pacific regions (according to WHO definitions of regions) reporting antenatal screening coverage for syphilis between 2010 and 2017, four countries reported coverage between 20% and 49% (India, Myanmar, Vanuatu, Papua New Guinea) and three reported coverage below 5% (Afghanistan, Indonesia, Solomon Islands) [11]. Yet unknowingly infected people can transmit infections to their sexual partners and infected women to their children through MTCT. This also prevents them from accessing

timely treatment leading to long-term complications that generate significant costs for the health system. In addition, low uptake of STIs screening services can exacerbate existing health disparities, with vulnerable populations, such as marginalised communities or migrant populations, facing additional barriers to accessing screening services.

To guide a path towards triple elimination of MTCT of HIV, syphilis, and hepatitis B in Asia and the Pacific, the WHO developed a regional framework [10]. This framework aims to eliminate these three infections in newborns and infants by 2030 in Asia. The key recommendations emphasise an integrated approach to triple elimination, recognising the interconnectedness of the three diseases and the potential for resource optimisation and highlights the importance of strengthening health systems to effectively deliver comprehensive services and achieve universal health coverage. The framework focuses on building capacity, improving laboratory and diagnostic services, ensuring a reliable supply chain for medicines and commodities, and improving reporting systems. It recognises the need for collaboration between different sectors beyond the health sector and the importance of sustainable financing mechanisms to support the implementation of elimination programmes. Meanwhile, it encourages the participation of women living with HIV, women affected by syphilis, and mothers with hepatitis B, men and communities in the design, implementation, and evaluation of programmes and policies.

Understanding barriers and facilitators influencing antenatal screening for STIs is essential to design effective screening interventions. The information will also be useful to help countries to achieve a key health target of the Sustainable Development Goals (SDGs), i.e., "end the epidemics of AIDS, tuberculosis, malaria and neglected tropical diseases and combat hepatitis, water-borne diseases, and other communicable diseases by 2030". A systematic review conducted by Blackstone et al. [12] investigated the barriers and facilitators to routine antenatal HIV screening in sub-Saharan Africa, using literature published between 2000 and 2015. They identified the fear of the screening results, perceived stigma towards HIV-positive people, fear of the partner's reaction in case of a positive test result, and perceived partner disapproval of the test as barriers to antenatal HIV screening. A high level of education, good knowledge of MTCT and HIV, and partner involvement in antenatal care were favourable factors for screening. Health system and provider issues affected the acceptance of antenatal screening. Good patient-provider communication, counselling to improve knowledge of pregnant women of the benefits of screening through counselling, and the perception that HIV screening is mandatory were facilitators to screening.

Barriers are likely to change over time, as societies evolve, beliefs change, or targeted interventions are put in place. There is no literature review summarising the evidence on barriers and facilitators to antenatal screening for HIV, syphilis, and hepatitis B in the Asian context. Factors affecting screening are likely to be different from those in the African context due to cultural and contextual differences. This hinders the development of targeted strategies and interventions to overcome barriers and improve the effectiveness of antenatal screening programmes. It also limits the application of the WHO framework towards triple elimination of MTCT of HIV, syphilis and hepatitis B. Health care providers in Asia may also lack guidance on how to effectively implement and improve antenatal screening programmes for STIs. Barriers preventing vulnerable communities from accessing screening are not known, which may contribute to disparities in health outcomes, with potentially negative impacts on maternal and child health.

In order to fill this evidence gap, this review aimed to investigate the barriers and facilitators to antenatal screening for HIV, syphilis, or hepatitis B for women in Asia. Its specific objectives were to identify available evidence and underline possible gaps in the research knowledge base surrounding this subject.

## Methods and analysis

The review and its reporting comply with the Preferred Reporting Items for Systematic Reviews and Meta-Analyses (PRISMA) checklist (S1 Table) and the protocol has been published on PROSPERO (registration number CRD42023435483).

### Search strategy

We conducted a comprehensive search of electronic databases including Ovid (MEDLINE, Embase, PsycINFO), Scopus, Global Index Medicus, and Web of Science was conducted to identify relevant studies published between 2000 and June 2023. The first search was conducted on 13 December 2021 and repeated on 10 June 2023 by LS. The keyword search was divided into five main groups: "barriers or facilitators", "antenatal screening", "HIV or syphilis or hepatitis B", and "Asian countries". The finalised search terms were developed through a trial-and-error process for use on Scopus and adapted to the different databases. The full key words used are shown in S1 File.

We used forward and backward citation searching to capture resources either citing or being cited by the included literature and searched the websites of the WHO, the World Bank and UNAIDS for reports.

### Inclusion criteria

The eligibility criteria for study inclusion were developed using the acronym SPIDER: S sample; P phenomenon of interest; D design; E evaluation; R research type [13] (Table 1).

### Study selection

Following the initial search, LS collated records and uploaded them into Rayyan [14] to facilitate screening. After removal of duplicates, two independent reviewers (LS and FM) screened titles and abstracts for relevance and assessed full text of potentially relevant article using the inclusion criteria. Those meeting inclusion criteria at full-text screen were included in our results. Any discrepancies were resolved through discussion or consultation with a third reviewer (NS) when needed.

### Data extraction

We used a standard form to extract key information including study characteristics (author, year, country, urban/rural setting, diseases considered), study design, sample, aim, identified significant barriers and facilitators to screening (e.g., odds ratios at the 95% confidence

**Table 1. Eligibility criteria for study inclusion.**

| | |
|---|---|
| **Sample** | Pregnant women or women of childbearing age in Asian countries, their family members, health workers and decision-makers (the search term including all the Asian countries as defined by the United Nations is provided in S1 File). |
| **Phenomenon of Interest** | Barriers and facilitators to antenatal screening and factors influencing screening uptake. Barriers were defined as factors discouraging or impeding screening uptake. Facilitators were defined as factors or resources enhancing screening uptake. Factors may also relate to the implementation and effectiveness of antenatal screening. |
| **Design** | Primary or secondary research studies, including quantitative, qualitative, and mixed-methods studies. |
| **Evaluation** | Antenatal screening programs or interventions related to the screening of HIV, syphilis, or hepatitis B during pregnancy. |
| **Research type** | Peer-reviewed journal articles in English conducted between 2000 and June 2023. |

interval, p-value < 0.05). We thematically analysed qualitative articles through an iterative process of reading and coding them using Andersen's framework [15]. This theoretical framework widely used in literature reviews on healthcare utilisation [16] provides understanding of how individuals and environmental factors influence health behaviours. The framework categorises predictors of health service use as i) Predisposing characteristics including demographic factors, social structure, and health beliefs that influence health services use. ii) Enabling factors allowing the individual to seek health services if needed. iii) Need factors including perceived needs of healthcare services use.

## Quality assessment

LS and FM assessed the quality of included studies using tools appropriate to the study design. The quality of the studies included was evaluated based on Von Elm et al's [17] checklist for observational studies and O'Brien et al's [18] checklist for qualitative studies. S2 and S3 Tables present the quality appraisal checklists for the considered studies. We scored each paper based on how many checklist items were met. Overall, papers that met over 75% of the checklist items were considered to be of high quality, those meeting 50% to 75% of the checklist were regarded as moderate quality, and those meeting less than 50% poor quality. Because the aim was to describe and synthesise a body of the literature and not determine an effect size, studies were not excluded based on quality.

## Data analysis and presentation

Descriptive characteristics of research studies were presented in tables. A narrative synthesis (Popay et al. 2006) was conducted to summarize the findings of the included studies. We did not combine quantitative estimates because of the heterogeneity of approaches and findings. Themes and patterns related to factors influencing screening uptake were identified and analysed and the final set of barriers and facilitators categorised according to Andersen [15]'s conceptual model.

# Results

After the selection process, 23 articles met the eligibility criteria and were included in the review. The PRISMA diagram provides an overview of the selection process (Fig 1).

## General study characteristics

Details about the articles included are presented in Table 2. Most included studies were on HIV screening, one was on syphilis screening [19], one on HIV and syphilis [20] and one on HIV, syphilis and hepatitis B [21]. Eight out of the 23 studies used data collected after 2015 [20, 22–28]. Six of the studies were conducted in Vietnam, five in India, three in Indonesia, two in Cambodia, and one each in Hong Kong, Mongolia, China, Afghanistan and Thailand. Nineteen of the studies (83%) used quantitative methods, three (15%) used qualitative methods, and one (2%) used mixed methods.

In the four studies that used qualitative methods, pregnant women were interviewed as well as other individuals such as health providers, district managers, husbands, and mothers. Sample sizes in quantitative studies ranged from 114 to 122,351 pregnant women, most often recruited during ANC visits. The quantitative studies were all cross-sectional except one from Indonesia, which was longitudinal [25]. Most quantitative studies used logistic regression models to determine the association between potential barriers and the outcome of interest.

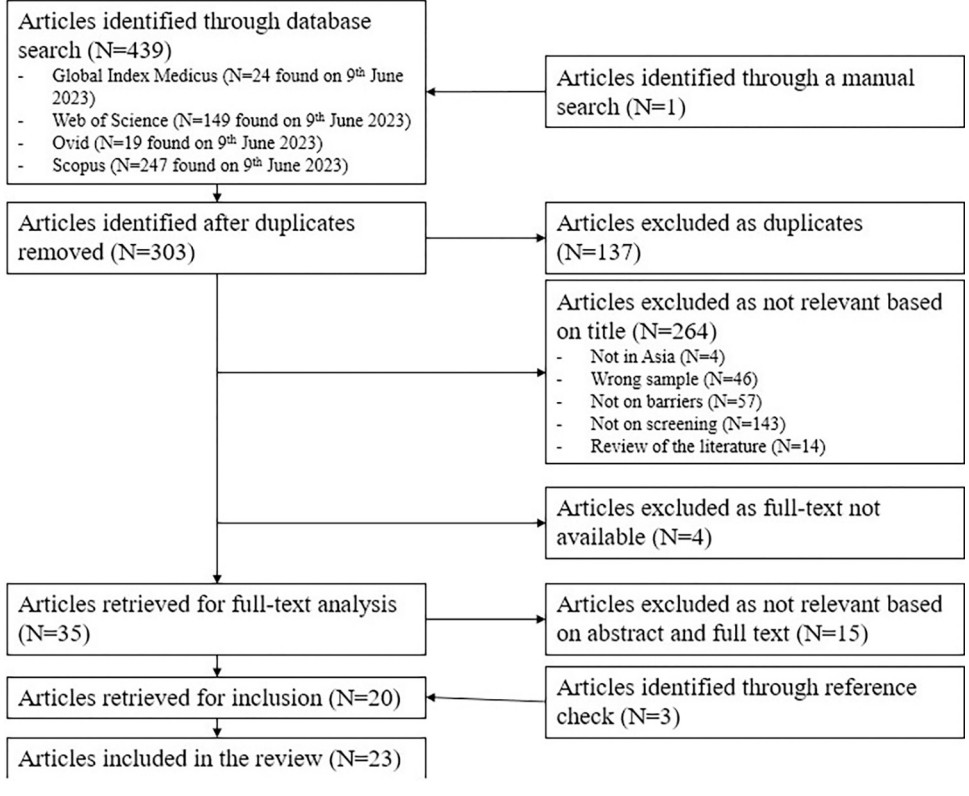

**Fig 1. PRISMA diagram of the research strategy.**

## Overviews of the barriers and the facilitators identified

The barriers and facilitators identified in the included articles are presented based on the categories of the Andersen's conceptual model (Table 3 and Fig 2).

**Predisposing characteristics.** Several predisposing characteristics were reported as either barriers or facilitators to antenatal screening for HIV and syphilis. In three studies conducted in Vietnam and India, age was associated with antenatal screening of HIV [22, 32, 33]. Pharris et al. [32] found that younger Vietnamese women were more likely to be screened while Bharucha et al. [33] found the opposite result in India. Khuu et al. [22] identified being younger than 30 years old as a barrier to antenatal screening.

Low education status of pregnant women was a barrier to antenatal screening in three studies conducted in Vietnam [22, 23, 29] and one in India [28]. Similarly, one study conducted in Hong Kong [39] and one in India [36] identified higher education as a facilitator to antenatal screening. However, the level of education associated with a positive likelihood of being screened varied between studies. For example, Khuu et al. [22] showed that nine or more years of education was associated with more acceptance of screening in Vietnam, whereas Sarin et al. [36] showed that this was true at more than six years of education in rural India.

Pregnant women's knowledge about HIV and PMTCT was associated with antenatal screening decisions. Lack of knowledge about HIV amongst pregnant women [28, 34, 36, 38], about the MTCT services [34], and about the availability of HIV testing facilities [35] were identified as barriers to screening in four studies in India, one in Cambodia and one in Thailand. Similarly, three studies conducted in Cambodia, Hong Kong and China found that a better knowledge of HIV amongst pregnant women was associated with a higher screening

**Table 2. Characteristics of selected papers.**

| Citation | Date | Country | Urban/rural | Disease | Sample | Study type | Aim |
|---|---|---|---|---|---|---|---|
| Dinh [29] | 2005 | Vietnam | Urban | HIV | 500 pregnant women 18 aged years and older who were first-time antenatal care (ANC) visitors and had never been tested or were unaware of their results | Quantitative | Identify the factors associated with declining HIV antenatal screening and the failure to return for results |
| Nguyen [30] | 2010 | Vietnam | Urban | HIV | 300 women who had recently delivered | Quantitative | Describe the uptake of antenatal HIV screening |
| Hạnh [31] | 2011 | Vietnam | Urban/rural | HIV | 1108 nursing mothers | Quantitative | Assess early uptake of HIV screening and the provision of HIV counselling among pregnant women |
| Pharris [32] | 2011 | Vietnam | Urban | HIV | 1108 pregnant women who attend antenatal care at primary and higher-level health facilities | Quantitative | Assess early uptake of HIV testing and the provision of HIV counselling among pregnant women |
| Khuu [22] | 2018 | Vietnam | Urban | HIV | 320 women who were tested during ANC | Quantitative | Identify reasons for late HIV screening among pregnant women |
| Chu [23] | 2019 | Vietnam | Urban/rural | HIV | 1484 women aged 15 to 49 years having a live birth within the last 2 years | Quantitative | Assess the socioeconomic inequalities in HIV screening during ANC |
| Bharucha [33] | 2005 | India | Urban | HIV | 6,702 pregnant women presenting in labour | Quantitative | Explore factors affecting the eligibility and acceptability of voluntary counselling and rapid HIV testing |
| Rogers [34] | 2006 | India | Rural | HIV | 202 pregnant women attending a rural ANC clinic | Quantitative | Investigate HIV-related knowledge, attitudes toward infant feeding practices, and perceived benefits and risks of HIV screening |
| Sinha [35] | 2008 | India | Rural | HIV | 400 women that have gave birth in the previous 12 months | Quantitative | Investigate HIV screening among rural women during pregnancy |
| Sarin [36] | 2013 | India | Rural | HIV | 357 women who had given birth in the last two years | Quantitative | Examine the prevalence and the barriers to HIV screening among pregnant women vulnerable to HIV due to their spouses' risky behaviours |
| Sharma [28] | 2022 | India | Urban/rural | HIV | 122,351 women aged 15–49 | Quantitative | Determine the factor associated with HIV screening during ANC |
| Lubis [24] | 2019 | Indonesia | Urban/rural | HIV | 20 private midwives | Qualitative | Examine midwives' perceptions of barriers and enabling factors about referring pregnant women for HIV screening |
| Wulandari [25] | 2019 | Indonesia | Urban/rural | HIV | 619 women to voluntary HIV counselling and screening clinics | Quantitative | Examine the rates of HIV screening uptake among pregnant women attending private midwife clinics |
| Baker [20] | 2020 | Indonesia | Rural | HIV, syphilis | 3382 pregnant women and 40 health workers involved in screening | Mixed-methods | Explore current practice, barriers and facilitators in the delivery of antenatal testing for anaemia, HIV and syphilis |
| Pakki [26] | 2020 | Indonesia | Rural | HIV | 42 health workers managers | Quantitative | Investigate the influence of training given to health workers on HIV testing uptake by pregnant women |
| Setiyawati [27] | 2021 | Indonesia | Urban | HIV | 350 housewives in districts that already implemented prevention mother-to-child transmission program | Quantitative | Assess the factors that influence the housewife attitude toward HIV testing |
| Kakimoto [37] | 2007 | Cambodia | Urban | HIV | 315 mothers who came to a childhood immunization with a child aged 6–24 months | Quantitative | Assess predictive determinants for HIV testing |
| Sasaki [38] | 2010 | Cambodia | Urban | HIV | 600 eligible mothers who were admitted to the hospital after delivery | Quantitative | Assess the prevalence of and barriers to HIV screening |
| Lee [39] | 2005 | Hong Kong | Urban | HIV | 3,500 pregnant women attending their first ANC visit | Quantitative | Investigate acceptance of universal HIV antibody screening programme |
| Munkhuu [19] | 2006 | Mongolia | Urban | Syphilis | 150 ANC providers and 27 senior doctors | Qualitative | Assess ANC providers' practices and opinions toward antenatal syphilis screening |

*(Continued)*

**Table 2.** (Continued)

| Citation | Date | Country | Urban/rural | Disease | Sample | Study type | Aim |
|---|---|---|---|---|---|---|---|
| Todd [21] | 2008 | Afghanistan | Urban | HIV, syphilis, hepatitis B | 114 doctors and midwives | Quantitative | Determine attitudes toward and utilization of testing for HIV, syphilis, and hepatitis B among obstetric care providers |
| Crozier [40] | 2013 | Thailand | Urban | HIV | 38 migrant pregnant women who had been through the HIV screening process 2013and 26 health personnel | Qualitative | Explore factors that relate to HIV screening decisions for migrant women |
| Li [41] | 2014 | China | Urban | HIV | 500 pregnant women recruited during their antenatal visit | Quantitative | Assess the prevalence of the willingness for HIV testing among pregnant women and cognitive factors associated with it |

uptake [37, 39, 41]. Moreover, Munkhuu et al. [19] found similar results for syphilis in their study conducted in Mongolia. Lack of knowledge about syphilis amongst pregnant women was associated with lower screening uptake. A study conducted in India [28] found that low exposure to mass media was associated with lower HIV screening uptake. Similarly in Hong Kong, Lee et al. [39] identified access to HIV information by means of posters, pamphlets, videos, and group talks as a facilitator to screening.

**Enabling factors.**   The role of enabling factors such as wealth, place of residence, husbands and health workers' roles, social and cultural norms or screening cost has been discussed in several articles.

Low household wealth or socio-economic status was a barrier even in countries where antenatal screening was free of charge. Three studies conducted in Mongolia, Vietnam, and India found low socio-economic status as being a barrier to antenatal screening for HIV [19, 23, 28]. Pharris et al. [32] identified higher economic status as a facilitator to antenatal screening for HIV in Vietnam.

Various studies have shown that the place of residence was associated with antenatal screening for HIV [22, 23, 25, 28, 30, 32, 33] and syphilis [19]. A study conducted in Vietnam [23] and another conducted in India [28] identified living in a rural area as a barrier to antenatal screening for HIV. Similarly, Wulandari et al. [25] and Pharris et al. [32] found that living in an urban area and a semi-urban area were facilitators to antenatal screening of HIV in Vietnam and Indonesia respectively. Proximity to the hospital is also a factor influencing antenatal screening uptake. Khuu et al. [22] and Nguyen, Christoffersen, and Rasch [30] found that living further away from the hospital (over 20km in the case of Khuu et al.) was a barrier to antenatal screening for HIV. Similar results were found by Munkhuu et al. [19] in Mongolia for the antenatal screening of syphilis. Meanwhile, Bharucha et al. [33] identified living closer to the hospital as a facilitator for antenatal screening of HIV in India.

Two studies conducted in Vietnam found a significant effect of occupation on the decision to be tested. For example, housewives, or labourers/farmers were less likely to be tested for HIV [22, 29]. Kakimoto et al. [37] identified high partner education level as a facilitator to antenatal screening in Cambodia. Meanwhile, Chu, Vo [23] found a negative association between belonging to ethnic minorities and being tested during pregnancy.

Several articles identified that their husband play a key role in women's decision to be screened. Fear of negative reactions from their husbands [34], husband's disapproval [29] and lack of support [40], and beliefs that their husbands have a bad attitude towards HIV testing [27] were identified as barriers to screening in India, Thailand, Indonesia and Vietnam respectively. Two studies conducted in Cambodia [37, 38] found that the perceived need to obtain partner's authorisation is a barrier to screening for HIV. Similar findings were found in

**Table 3. Barriers and facilitators to antenatal screening for HIV, syphilis and hepatitis B identified in the selected papers based on the Andersen's conceptual model.**

| Citation | Date | Country | Diseases | Predisposing characteristics | Enabling factors | Need factors |
|---|---|---|---|---|---|---|
| Bharucha [33] | 2005 | India | HIV | Facilitators:<br>• Being older<br>• Living closer to the hospital | Barriers:<br>• Being too far along in the birth delivery process when the opportunity to test arises<br>Facilitators:<br>• Having had antenatal care in the hospital rather than in other health facilities | |
| Dinh [29] | 2005 | Vietnam | HIV | Barriers:<br>• Being a housewife<br>• Low level of education | Barriers:<br>• Fear of husband's disapproval<br>• Perception of poor healthcare availability | Barriers:<br>• Low-risk perception |
| Lee [39] | 2005 | Hong Kong | HIV | Facilitators:<br>• High level of education<br>• Good HIV knowledge<br>• Access to HIV information by means of posters, pamphlets, videos and group talks | Facilitators:<br>• Healthcare workers' recommendations to be screened | Barriers:<br>• No or low-risk perception<br>Facilitators:<br>• Good perceived benefits of screening |
| Rogers [34] | 2006 | India | HIV | Barriers:<br>• Low knowledge of HIV | Barriers:<br>• Fear of negative reactions from husbands, parents, and community<br>• Fear of stigma and discrimination | |
| Munkhuu [19] | 2006 | Mongolia | Syphilis | Barriers:<br>• Low knowledge of syphilis<br>• Being poor<br>• Long travel distance to get tested | Barriers:<br>• Limited time for screening due to antenatal visits starting late in pregnancy<br>• Complexity of testing service system<br>• Undersupplied screening materials<br>• Healthcare workers not in favour of screening | Barriers:<br>• Reporting previous sexually transmitted diseases |
| Kakimoto [37] | 2007 | Cambodia | HIV | Facilitators:<br>• Basic knowledge of HIV transmission<br>• High partner education level | Barriers:<br>• Need to obtain husband's approval to be tested | |
| Sinha [35] | 2008 | India | HIV | Barriers:<br>• Low awareness of existing HIV testing facilities | Barriers:<br>• Never received HIV counselling before | |
| Todd [21] | 2008 | Afghanistan | HIV, syphilis, hepatitis B | | Facilitators:<br>• High acceptance of screening by providers<br>Barriers:<br>• Providers' perceptions that infections were rare<br>• Provider's low perceived likelihood of infection based on healthy appearance<br>• Stigma toward infected individuals<br>• Need to obtain husband's approval to be tested | |
| Nguyen [30] | 2010 | Vietnam | HIV | Barriers:<br>• High distance to the hospital | | |
| Sasaki [38] | 2010 | Cambodia | HIV | Barriers:<br>• Low knowledge of HIV | Barriers:<br>• Lack of access to antenatal care services<br>• Need to obtain husband's approval to be tested | |
| Hạnh [31] | 2011 | Vietnam | HIV | | Facilitators:<br>• First antenatal check-up at primary health facilities rather than at district and provincial health facilities | |
| Pharris [32] | 2011 | Vietnam | HIV | Facilitators:<br>• Younger age<br>• Residence in a semi-urban area<br>• Higher economic status | | Barriers:<br>• Low perception of risk |

(*Continued*)

**Table 3.** (Continued)

| Citation | Date | Country | Diseases | Predisposing characteristics | Enabling factors | Need factors |
|---|---|---|---|---|---|---|
| Crozier [40] | 2013 | Thailand | HIV | Barriers:<br>• Low knowledge of HIV and mother-to-child transmission | Barriers:<br>• Language differences between health worker and pregnant women<br>• Concern about the reactions of health workers<br>• Financial barriers<br>• Costs and time of transportation<br>• Provider's lack of time to inform women properly<br>• Having only one antenatal check-up<br>• Lack of support from husband | Barriers:<br>• Low perception of risk |
| Sarin [36] | 2013 | India | HIV | Facilitators:<br>• More than six years of education<br>• Good knowledge of HIV | Facilitators:<br>• Discussions with husband about HIV<br>• Seeking antenatal care in government district hospitals and private clinics as opposed to community health centres (not equipped with either HIV counselling or testing facilities) | |
| Li [41] | 2014 | China | HIV | Facilitators:<br>• Good knowledge of HIV | Facilitators:<br>• Less perception of social stigma | Facilitators:<br>• High perception of risk |
| Khuu [22] | 2018 | Vietnam | HIV | Barriers:<br>• Younger than 30 years old<br>• Nine or fewer years of education<br>• Working as a homemaker or worker/farmer<br>• Living 20km or more from the hospital | Barriers:<br>• Having received antenatal care at private clinic/hospital only | Barriers:<br>• Low perceived benefits of screening |
| Chu [23] | 2019 | Vietnam | HIV | Barriers:<br>• Belonging to ethnic minorities<br>• Having primary or less education<br>• Being poor<br>• Living in rural areas | | |
| Lubis [24] | 2019 | Indonesia | HIV | | Facilitators:<br>• Free HIV screening<br>• Reward and punishment system to motivate providers<br>• Training for health workers<br>Barriers:<br>• Fear of stigma<br>• Limited voluntary counselling and testing opening hours do not cater for those in employment<br>• Not a one-roof for ANC and VCT services<br>• Providers disguising or not revealing purpose of the blood testing for fear of causing offense | |
| Wulandari [25] | 2019 | Indonesia | HIV | Facilitators:<br>• Living in urban area | | |
| Baker [20] | 2020 | Indonesia | HIV, syphilis | | Barriers:<br>• National policy on testing not widely disseminated<br>• Testing not seen as a priority intervention<br>• Multiple small-scale funding sources<br>• Tests seen as expensive by pregnant women<br>• Lack of knowledge and training of providers<br>• Shortage of laboratory personnel<br>• Shortage of tests and laboratory resources<br>• Stigma amongst providers and community<br>• Lack of time from pregnant women<br>• Fear of the results | Barriers:<br>• Perceived low prevalence |
| Pakki [26] | 2020 | Indonesia | HIV | | Facilitators:<br>• Health workers training on predisposing factors of provider-initiated testing and counselling of HIV | |

(*Continued*)

**Table 3.** (Continued)

| Citation | Date | Country | Diseases | Predisposing characteristics | Enabling factors | Need factors |
|---|---|---|---|---|---|---|
| Setiyawati [27] | 2021 | Indonesia | HIV | | Barriers:<br>• Pregnant women's beliefs that their husbands have a bad attitude towards HIV testing | Barriers:<br>• Low perceived benefits of screening |
| Sharma [28] | 2022 | India | HIV | Barriers:<br>• Low educational level<br>• Low knowledge of HIV<br>• Being poor<br>• Living in rural area<br>• Low exposure to mass media | | |

Afghanistan by Todd et al. [21] for antenatal screening of syphilis and hepatitis B. Similarly, Sarin et al. [36] reported that having discussions with spouses about HIV in India encouraged women's screening for HIV.

Various studies have shown that social and cultural factors were key barriers to antenatal screening for HIV, syphilis or hepatitis B. Todd et al. [21] identified stigma toward infected people as a barrier to antenatal screening for HIV, syphilis, and hepatitis B in Afghanistan. Similar results were found by Baker et al. [20] in Indonesia for the screening of HIV and syphilis, and Lubis et al. [24] and Rogers et al. [34] for the screening of HIV. This last article also identified the fear of negative reactions from parents and community as a barrier. Similarly, Li et al. [41] found that lower perception of social stigma was associated with higher screening uptake.

Time was also associated with antenatal screening decisions for HIV and syphilis. It was a barrier both from the supply and the demand side. Working pregnant women reported that limited opening hours of screening centres were a major health-facility related barrier to antenatal screening for HIV in Indonesia [24]. Limited time to inform women properly about HIV during pregnancy and antenatal screening [40] as well as limited time to perform screening for syphilis [19] were barriers to antenatal screening in Thailand and Mongolia. From the demand side, long travel time to access antenatal screening services was associated with lower HIV screening uptake in Thailand [40]. Similarly, lack of time was identified as a barrier to screening for HIV and syphilis in Indonesia by Baker et al. [20]. Meanwhile, Bharucha et al. [33] found that being offered testing too late in pregnancy as associated with lower screening uptake for HIV.

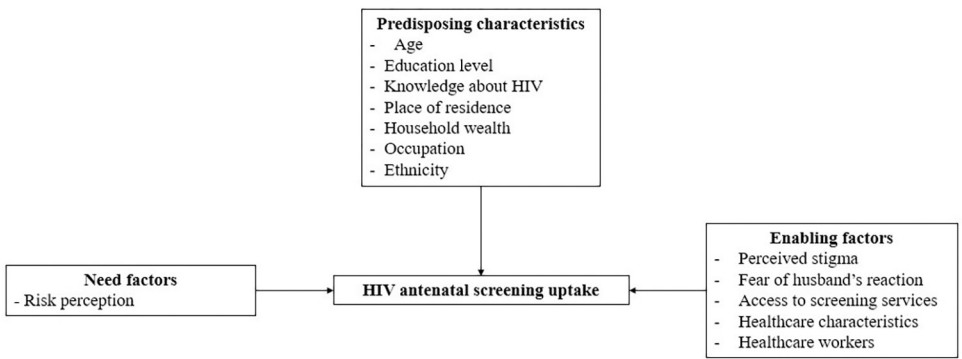

**Fig 2. Flowchart of factors influencing antenatal screening for HIV, syphilis and hepatitis B based on the Andersen's conceptual model.**

The type of screening provider was a factor associated with screening in various studies. Hạnh, Gammeltoft, and Rasch [31] showed that, in Vietnam, having the first antenatal check-up at a commune health station was a factor associated with an increased probability of being tested, compared with district and provincial health facilities. Similarly and in the same country, having received ANC only at a private clinic/hospital was found to be a barrier [22]. However, in India, Sarin et al. [36] found that seeking ANC at government district hospitals and private clinics, as opposed to community health centres not equipped with either HIV counselling or testing facilities, had a positive effect on the probability of receiving HIV screening. Similar results were found by Bharuch et al. [33] in India. Some facilities lack screening materials and this was associated with lower screening of syphilis in Mongolia [19] and lower screening of HIV and syphilis in Indonesia [20]. In addition, a study carried out in Indonesia [24] revealed that the lack of antenatal care and screening services in the same building was a barrier to HIV screening. In Cambodia, the lack of access to ANC services outside the capital city was a barrier to screening for HIV [38].

Healthcare workers play a key role in screening decisions. In Vietnam, Dinh, Detels and Nguyen [29] found that a poor perception of healthcare availability was negatively associated with screening for HIV. Fear that healthcare workers would become impatient with them or that their questions would not be considered important was a barrier in Thailand [40], and concern that healthcare workers were opposed to antenatal screening for syphilis impeded testing in Mongolia [19]. Similarly, Lee et al. [39] identified health worker recommending HIV testing as a facilitator of screening. A study conducted in Vietnam [32] identified never having received antenatal HIV counselling as a barrier to screening and another identified a language barrier between health workers and women as barriers [40]. High acceptance of screening for HIV, syphilis and hepatitis B was also a factor increasing screening uptake in Afghanistan [21]. Pakki et al. [26] and Lubis et al. [24] found that, in Indonesia, health worker training as well as reward and punishment system to motivate them was associated with higher antenatal HIV screening. This is consistent with findings reported in Indonesia for HIV and syphilis screening [20]. Todd et al. [21] found that provider perceptions of low infection rates and assumptions on a person's likelihood of infection based on a healthy appearance were associated with lower screening uptake of HIV, syphilis and hepatitis B in Afghanistan. Baker et al. [20] also identified shortage of laboratory personnel as a barrier to screening.

Costs of screening was also identified as factor influencing HIV and syphilis screening uptake. Tests being seen as expensive by pregnant women was identified as a barrier to HIV and syphilis screening in Indonesia [20]. Similarly, Crozier et al. [40] found that costs of screening and transportation represent barriers to screening of HIV and syphilis in Thailand.

At the national-level, enabling factors were identified by two studies in Mongolia and Indonesia [19, 20]. Munkhuu et al. [19] identified the complexity of the syphilis testing service system as a barrier to antenatal screening. Similarly, Baker et al. [20] found that poor dissemination of national policy on screening, not seeing screening as a priority intervention, and funding consisting of multiple small-scale sources were barriers to HIV and syphilis screening in Indonesia.

Finally, Crozier, Chotiga et Pfeil [40] showed that having only one ANC check-up was associated with low screening uptake.

**Need factors.** Few need factors were identified as barriers or facilitators in antenatal screening for HIV and syphilis. Four studies conducted in Hong Kong, Vietnam and Thailand found that low perceived risk of HIV was associated with low screening [29, 32, 39, 40]. Similarly, Lee, Yang, and Kong [41] found that, in China, high perceived risk of HIV was associated with high screening. In a study investigating barriers and facilitators in the delivery of antenatal testing for anaemia, HIV, and syphilis, Baker et al. [20] identified perceived low prevalence

of HIV and syphilis as barriers to antenatal screening in Indonesia. Two studies found that believing that HIV testing was not important during pregnancy was associated with a lower screening uptake in Indonesia and Vietnam [22, 27]. Similar Lee et al. [39] identified the perception of the benefits of HIV screening as a factor facilitating it. Finally, Munkhuu et al. [19] found that women who previously reported STIs were less likely to be screened in Mongolia.

## Discussion

This study is the first to provide a narrative synthesis of the current literature on barriers and facilitators to antenatal screening for HIV, syphilis and hepatitis B in Asia. This systematic review of qualitative, quantitative and mixed-method studies shows that there are research gaps into the factors influencing screening for syphilis and hepatitis B, with most of the studies reviewed focusing on HIV. This review therefore effectively allows conclusions to be drawn about HIV alone.

Antenatal screening for HIV in Asia is influenced by a range of factors including predisposing characteristics (age, education level, wealth, place of residence, knowledge about HIV), enabling factors (husband support, health facilities characteristics, health workers' support and training) and need factors (risk perception, perceived benefits of screening). These factors are similar to those identified in a review conducted by Blackstone et al. [12] in sub-Saharan Africa. In our literature review, as in the sub-Saharan African context, being better-off and highly educated were identified as facilitators. In both contexts, pregnant women's lack of knowledge about HIV appears to be a significant barrier to antenatal HIV screening. Our results suggest that antenatal screening could be improved by facilitating access to information for women, their husbands and health workers. Most studies have emphasised the importance of improving dissemination of information about HIV and HIV testing in order to improve uptake of antenatal screening. Unlike Blackstone et al.'s review of the literature in the sub-Saharan African context [12], our review did not identify fear of results as such as a barrier to testing, but more broadly fear of partner reactions and potential violence in the event of a positive result. We did not find that cultural gender norms to be barrier, such as "testing is a woman's business", as found by Blackstone et al. [12]. However, women in this review mentioned the need to obtain a husband's approval to undergo screening. In both African and Asian contexts, societal stigma towards HIV-positive people proved to be a major barrier to HIV testing. Our findings, and those of Blackstone et al. [12], suggest that antenatal screening could be improved by strengthening the health care system. Both reviews highlighted the role of healthcare and communication professionals in increasing antenatal screening rates. In the sub-Saharan African context the perception of screening being mandatory was a barrier to screening, but this did not emerge in our literature review.

Although the studies we reviewed were all conducted in Asia, they spanned very different contexts. It is reasonable to assume that the barriers to antenatal screening will differ between Hong Kong and India for instance. Guidelines about screening and adherence to guidelines differ between countries. A review of maternal health care policies in eight countries in the Western Pacific region [42] found that WHO recommendations on antenatal HIV screening were not included in antenatal care guidelines in two countries. In 2018, 37 countries in the Asia Pacific region promoted antiretroviral therapy for all pregnant and breastfeeding women living with HIV, but in six of these countries, the policy is being implemented in less than 50% of all maternal and child health sites [43]. Reported barriers in the Hong Kong study were mainly focused on the demand side [39], whereas the Mongolia study identified many supply-side barriers [19]. This highlights the need for qualitative studies in Asian contexts to investigate context-dependent factors that may be missed in quantitative studies.

As stigmatisation of people with STDs is one of the main factors preventing pregnant women from being screened, interventions should provide information and counselling to pregnant women and their husbands, tailored to low-literacy populations to help reduce stigma and increase uptake [36, 38, 39]. Raising awareness within communities of the importance of male partner involvement, the benefits of screening and adherence to treatment could increase demand for antenatal screening services. However, studies on awareness campaigns about HIV in Vietnam [44] and Thailand [45] showed that the stigma attached to social judgement is difficult to reduce. Various studies recommended the integration of HIV screening into community level ANC services [23, 25, 30, 31, 39] and the development of opt-out approaches for those who prefer not to test [29, 35], as recommended in sub-Saharan Africa by Blackstone et al. [12]. We found that husbands play a key role in encouraging pregnant women to undergo screening. Interventions to improve husbands' knowledge and involvement in maternal and newborn health had a positive impact on maternal health behaviour in Bangladesh [46] and Nepal [47]. To reduce social and financial barriers to antenatal screening, screening should be offered to pregnant women universally free of cost [32, 39]. Currently, national budgets do not cover all the costs associated with antenatal screening in all Asian countries. In the 17 Asian countries for which data on the cost of screening pregnant women for HIV, syphilis and hepatitis B were available in 2017, HIV screening of pregnant women was free in all of these countries, syphilis screening in 14 countries and hepatitis B screening was free in eight countries [11]. Finally, the quality of services depends on the availability and capacity of healthcare workers. To reduce the persistence of inappropriate healthcare practices in pregnancy, interventions need to develop health worker training programmes on STIs and pregnancy screening. A successful initiative in Cambodia in decreasing risky sexual intercourse and improving the access to sexual and reproductive health care services has focused on training community health workers in sexual and reproductive, maternal, neonatal, child and adolescent health [48].

Adolescent pregnancy is still common in the region with 3.7 million births to adolescent girls aged 15–19 every year in Asia and the Pacific [49]. Pregnant adolescents are very vulnerable and are known to have poor outcomes for both mother and child [50]. This systematic review of the literature highlighted a lack of age-specific data, particularly in relation to adolescent pregnancy, and confirmed the need to fill this research gap. Similarly, a systematic literature review of interventions addressing health outcomes for pregnant adolescents in low- and middle-income countries highlighted the need to develop studies to design high-quality care and services for pregnant adolescents [51].

Several limitations to this study should be noted. Firstly, most studies sampled pregnant women through ANC services. However, women who have not sought ANC may face the greatest barriers to testing. Due to resource constraints, only articles in English were reviewed, which may limit access to the grey literature and studies published in other languages (especially Chinese). Finally, different studies were undertaken in different contexts and using different methods. This heterogeneity limits our ability to compare between studies. However, this systematic review follows a rigorous method of article selection and analysis. It complements existing literature reviews on barriers to antenatal screening, particularly in sub-Saharan Africa [12, 52].

## Conclusion

The main barriers to antenatal screening in this systematic review were stigmatisation of infected individuals, lack of involvement of husbands and healthcare system factors. To improve uptake of antenatal screening interventions to improve community and husband

involvement, awareness campaigns with communities and health workers, and training of health workers on STI issues are needed. While countries vary in their contexts and implementation of international recommendations on integrated antenatal screening for STIs, in all settings the planning, implementation, reporting and monitoring of interventions to eliminate mother-to-child transmission require coordination between different health system stakeholders at national, regional and local levels to avoid gaps or duplication. Global, regional and national guidelines need to be harmonised to avoid gaps and duplication between disease-specific and maternal and child health programs and guidelines. Integration of services for different diseases should be prioritised where possible. However, studies to examine the barriers and facilitators to antenatal screening for syphilis and hepatitis B and to examine the behavioural determinants of antenatal screening in Asia are still needed.

## Supporting information

**S1 Table. The Preferred Reporting Items for Systematic Reviews and Meta-Analyses (PRISMA) checklist.**
(DOCX)

**S2 Table. Quality appraisal checklists of included qualitative studies based on O'Brien, Harris et al. (2014)'s checklist.**
(DOCX)

**S3 Table. Quality appraisal checklists of included quantitative studies based on Von Elm, Altman et al. (2007)'s checklist.**
(DOCX)

**S1 File. Query performed on Scopus on 10 June 2023.**
(DOCX)

## Author Contributions

**Formal analysis:** Lucie Sabin.

**Investigation:** Lucie Sabin, Faith Miller.

**Methodology:** Lucie Sabin, Hassan Haghparast-Bidgoli, Faith Miller, Naomi Saville.

**Supervision:** Hassan Haghparast-Bidgoli, Naomi Saville.

**Validation:** Faith Miller, Naomi Saville.

**Writing – original draft:** Lucie Sabin.

**Writing – review & editing:** Lucie Sabin, Hassan Haghparast-Bidgoli, Faith Miller, Naomi Saville.

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
