## [Decision Letter · Decision Letter 0]

19 May 2023

PONE-D-23-03027Barriers and facilitators to antenatal screening for sexually transmitted diseases in Asia: a scoping review.PLOS ONE

Dear Dr. Sabin,

Thank you for submitting your manuscript to PLOS ONE. After careful consideration, we feel that it has merit but does not fully meet PLOS ONE’s publication criteria as it currently stands. Therefore, we invite you to submit a revised version of the manuscript that addresses the points raised during the review process.

Carefully consider both reviewers' comments especially with regards to the framework that was used for the study and also labelling of the study as a scoping review. The various components and processes that are necessary for a scoping review ought to be presented in the manuscript.

We look forward to receiving your revised manuscript.

Kind regards,

Gifty Dufie Ampofo, M.D., Ph.D

Academic Editor

PLOS ONE

2. Please include a copy of Table 4 and 5 which you refer to in your text on page 6.

Reviewers' comments:

Reviewer's Responses to Questions

**Comments to the Author**

1. Is the manuscript technically sound, and do the data support the conclusions?

Reviewer #1: No

Reviewer #2: No

2. Has the statistical analysis been performed appropriately and rigorously? 

Reviewer #1: N/A

Reviewer #2: N/A

3. Have the authors made all data underlying the findings in their manuscript fully available?

Reviewer #1: Yes

Reviewer #2: Yes

4. Is the manuscript presented in an intelligible fashion and written in standard English?

Reviewer #1: No

Reviewer #2: Yes

5. Review Comments to the Author

Reviewer #1: The topic is of great interest, however the manuscript seems hurriedly put together. Concepts involved such as Andersen's model are not clearly and coherently applied in the course of the write-up.

The title for example does not capture the full essence of the review - it seems to be based on the "regional framework for the triple elimination of MTCT of HIV, Hep B and syphillis in Asia and Pacific" (lines 99 - 101)- that should have reflected clearly in the title. Such an understanding should have been provided briefly in the "Background" section of the script.

Secondly, the target population of the review was not clear right from the "title". Who did the review focus on - pregnant women, healthcare providers, relatives, etc. or all of them? That is not clear from the manuscript.

The objectives of the review as stated in lines 74 and 75 are not meant for a scoping review, but some other approaches that would generate clear evidence such as a review of qualitative evidence.

The methods section has lots of lapses.

a. A key concept such as the "population of the review" is not defined. The "search strategy" should have been systematically presented - e.g., keywords/ subject headings/ index terms that were used, example of a search strategy of at least one of the databases should have been placed in the appendices; the search should have been as exhaustive as possible - searching two databases does not sound exhaustive enough.

b. The "inclusion and exclusion criteria" should have been clearly defined each criterion - this subsection at best is confusing to the reader as it does not serve its purpose of clearly pointing out how studies were included in the review. Studies of experimental designs were excluded with no apparent reasons as to why they were.

c. Under "Data extraction", the Andersen's framework was said to have been the guide - it would have been great to have this clearly "tabulated" with "findings" related to each component of the framework duely and systematically presented. The quality of studies was assessed, but this outcome did not feature further into any decision-making or discussion as to how study quality influenced the review process.

d. Andersen's model as defined in lines 123 - 124 does not seem to be in sync with the cited publication (11), i.e., predisposing, enabling, and needs factors. "Enabling factors" were not given the prominence required.

Aside the mention of "facilitators" on line 147, this key component of the review was hardly addressed. Furthermore in the results section, the nature of the findings as were presented did not necessarily aptly fit the defined factors under the Andersen model.

In its current form, this manuscript in my view is not fit for publication.

Reviewer #2: Review Comments on manuscript PONE-D-23-03027

Title

…..perhaps the title should be restricted to HIV, Syphilis and Hepatitis B rather than ‘sexually transmitted diseases’

Introduction

Despite being a scoping review, a little more detail will improve the Background and situate arguments in better context.

• Lines 50/51…….the authors can provide recent data on the morbidity and mortality they refer to…..first globally and in the Asian context

Quantify the prevalence of these STDs in Asia (at least present data from some Asian countries)

• Lines 58/61…..can the authors assign, at least, an estimate of how many children are born to these STDs? Can they quantify antenatal screening for STDs in Asia? Can they give us an idea of how low is ‘low’.

I think it would be useful to give a brief overview of this WHO regional framework and what different prescriptions it gave compared to whatever existed before its formation

• Lines 68/70……The authors may want to give more meaning to the listed ‘barriers’ and how they relate to uptake of HIV screening services.

It would be particularly interesting to see what the story is for “health system and health care provider issues”

In many jurisdictions in sub-Saharan Africa, HIV screening is part of the antenatal care package and is offered using the opt-out model.

• There is no literature review summarising the research-based evidence on barriers and facilitators to antenatal screening for STDs in the Asian context.

Why is the statement above a problem? What is the burden of maternal and child morbidity and mortality in Asia in the context of HIV, Syphilis, Hepatitis C? What are the fall-outs from the supposed low uptake of available screening services? What do we stand to lose if this review is not done to better understand barriers and facilitators that can help inform useful interventions?

• Can the authors rewrite the Methods section without ‘We’?

• Line 84. We used a very inclusive search strategy to ensure that no item was missed…..Can you give a 100% guarantee no item was missed?

• There appears to be something in Line 92 that is not supposed to be there

Inclusion and Exclusion Criteria

The inclusion and exclusion criteria appear to be ‘all over the place’. They can be made more focused.

We did not include studies investigating antenatal screening for other STDs……..This does not qualify as an exclusion criterion because you have earlier specified that you are dealing with HIV, Hep B and Syphilis

• The data extraction sub-section has nothing on “facilitators”

• Line 121…..the authors may want to justify the choice of Andersen’s conceptual model over other models they could have used.

• For Table 1, I think the year the study was conducted ought to be in separate column by itself. This will enable readers to relate them to implementation of the MDGs and contextualize them.

• I have some questions relating to Fig.1;

……you mentioned Google Scholar but I don’t see in the flow diagram

…….I am struggling to understand how you excluded 546 articles because the studies were not conducted in Asia when ‘Asia’ should have been a key part of your search strategy. Kindly elaborate on how this happened.

• Lines 210/211…….please give more meaning to ‘perceptions of poor healthcare support’ and ‘concerns about the reactions of healthcare workers

Secondly, of the 16 articles, 15 were on HIV and 1 was on Syphilis with nothing on Hepatitis B. In this context, I fail to see how you can make any reasonable pronouncements on Syphilis and Hepatitis B screening uptake.

On this basis, I suggest you drop off Syphilis and Hep B and make your work entirely about uptake of HIV screening than about STDs and appropriately reorient your discussion to that effect.

I look forward to reading a new version of your work.

6. PLOS authors have the option to publish the peer review history of their article (what does this mean?). If published, this will include your full peer review and any attached files.

Reviewer #1: **Yes: **Yeetey ENUAMEH

Reviewer #2: No

---

## [Author Response · Author response to Decision Letter 0]

5 Aug 2023

Dear Doctor Ampofo,

We thank you and the reviewer for your time and generous comments provided on our manuscript PLONE-D-23-03027 entitled "Barriers and facilitators to antenatal screening for sexually transmitted diseases in Asia: a scoping review", and we thank you for the opportunity to address these comments. After consideration of your comments, we have made several improvements. 

The comments provided are shown in bold below, with our responses in italics. 

Editor's Comments to the Author:

Response: We have ensured that the manuscript meets the stylistic requirements of PLOS ONE.

2. Please include a copy of Tables 4 and 5 which you refer to in your text on page 6.

Response: A copy of tables 4 and 5 has been included under the name S2 Table and S3 Table.

3. Please include captions for your Supporting Information files at the end of your manuscript, and update any in-text citations to match accordingly.

Response: Captions for our Supporting Information files have been included at the end of the manuscript.

Reviewers’ Comments to the Author:

Reviewer #1:

The topic is of great interest, however the manuscript seems hurriedly put together. Concepts involved such as Andersen's model are not clearly and coherently applied in the course of the write-up. 

Response: Thank you for your comment. We now define Andersen’s model (lines 148-153) and clearly and systematically apply the concepts cited. We have added Table 3 and systematically structured our results according to each component of Andersen’s model. The "facilitators" have been taken into account appropriately in the review.

The title for example does not capture the full essence of the review - it seems to be based on the "regional framework for the triple elimination of MTCT of HIV, Hep B and syphilis in Asia and Pacific" (lines 99 - 101)- that should have reflected clearly in the title. Such an understanding should have been provided briefly in the "Background" section of the script.

Secondly, the target population of the review was not clear right from the "title". Who did the review focus on - pregnant women, healthcare providers, relatives, etc. or all of them? That is not clear from the manuscript.

Response: We have changed the initial title from “Barriers and facilitators to antenatal screening for sexually transmitted diseases in Asia: a scoping review” to “A systematic review of barriers and facilitators to antenatal screening for HIV, syphilis or hepatitis B in Asia: perspectives of pregnant women, their relatives and health care providers.” The role of the WHO framework as the basis for this study has also been clarified in the background section (lines 76-89).

The objectives of the review as stated in lines 74 and 75 are not meant for a scoping review, but some other approaches that would generate clear evidence such as a review of qualitative evidence.

Response: On the basis of all the reviewers' comments, we decided to carry out a systematic analysis of the literature rather than a scoping review, using a second person to screen and review articles in duplicate and by searching other databases. We adapted the objectives to those of a systematic literature review (lines 113-115).

The methods section has lots of lapses.

a. A key concept such as the "population of the review" is not defined. The "search strategy" should have been systematically presented - e.g., keywords/ subject headings/ index terms that were used, an example of a search strategy of at least one of the databases should have been placed in the appendices; the search should have been as exhaustive as possible - searching two databases does not sound exhaustive enough.

Response: Thank you for this comment. To make the search as exhaustive as possible and in line with other reviews of this nature, we searched Ovid (MEDLINE, Embase, PsycINFO), Scopus, Global Index Medicus and Web of Science. Our database search strategy has been included in S1 File.

b. The "inclusion and exclusion criteria" should have been clearly defined each criterion - this subsection at best is confusing to the reader as it does not serve its purpose of clearly pointing out how studies were included in the review. Studies of experimental designs were excluded with no apparent reasons as to why they were.

Response: The eligibility criteria for the inclusion of studies have been rewritten and tabulated in Table 2 using the acronym SPlDER: S sample; P phenomenon of interest; D design; E evaluation; R research type. Our original inclusion/exclusion criteria were unchanged except that we also decided to include experimental design studies.

c. Under "Data extraction", the Andersen's framework was said to have been the guide - it would have been great to have this clearly "tabulated" with "findings" related to each component of the framework duly and systematically presented. The quality of studies was assessed, but this outcome did not feature further into any decision-making or discussion as to how study quality influenced the review process. 

Response: We agree that a more systematic application of Andersen’s framework improves the paper so have rewritten the results section to clearly reflect each of the elements of the Andersen framework and added Table 3. As the aim of the review was to describe and synthesise a body of literature and not to determine effect size, we did not exclude studies on the basis of their quality assessment (lines 160-161) but we have summarised the quality assessment in S2 and S3 Tables to enable readers to see the quality of the evidence included in the review. 

d. Andersen's model as defined in lines 123 - 124 does not seem to be in sync with the cited publication (11), i.e., predisposing, enabling, and needs factors. "Enabling factors" were not given the prominence required. 

Response: We agree with this comment and have now given more explanation of the ‘enabling factors’ of antenatal screening (lines 151-152).

Aside the mention of "facilitators" on line 147, this key component of the review was hardly addressed. Furthermore in the results section, the nature of the findings as were presented did not necessarily aptly fit the defined factors under the Andersen model. 

Response: As explained above we have now explicitly defined the "facilitators" in Table 1 and entirely rewritten the results section to follow the factors defined in Andersen’s model.

In its current form, this manuscript in my view is not fit for publication.

Response: Thank you for your detailed comments. We hope that the changes you have made will enable the study to be published.

Reviewer #2:

Title…..perhaps the title should be restricted to HIV, Syphilis and Hepatitis B rather than ‘sexually transmitted diseases’.

Response: Thank you. We have changed the title to “A systematic review of barriers and facilitators to antenatal screening for HIV, syphilis or hepatitis B in Asia: perspectives of pregnant women, their relatives and health care providers.”

Introduction: Despite being a scoping review, a little more detail will improve the Background and situate arguments in better context.

Response: We have expanded the introduction to better place the study in context. We have also changed from a scoping to a systematic review.

Lines 50/51…….the authors can provide recent data on the morbidity and mortality they refer to…..first globally and in the Asian context Quantify the prevalence of these STDs in Asia (at least present data from some Asian countries).

Response: Data on mother-to-child transmission of HIV, syphilis and hepatitis B were added to the introduction. The prevalence of the STDs considered was quantified in the introduction section.

Lines 58/61…..can the authors assign, at least, an estimate of how many children are born to these STDs? Can they quantify antenatal screening for STDs in Asia? Can they give us an idea of how low is ‘low’.

Response: We have added an estimate of the number of children born with these STDs (lines 54-59), as well as a quantification of antenatal screening for STDs in Asia (lines 63-68).

I think it would be useful to give a brief overview of this WHO regional framework and what different prescriptions it gave compared to whatever existed before its formation

Response: Thank you for this suggestion. We have added an overview of the WHO regional framework and its main prescriptions (lines 76-87).

Lines 68/70……The authors may want to give more meaning to the listed ‘barriers’ and how they relate to uptake of HIV screening services. It would be particularly interesting to see what the story is for “health system and health care provider issues”. In many jurisdictions in sub-Saharan Africa, HIV screening is part of the antenatal care package and is offered using the opt-out model.

Response: Thank you for this suggestion. The obstacles listed have been detailed for greater clarity.

There is no literature review summarising the research-based evidence on barriers and facilitators to antenatal screening for STDs in the Asian context. Why is the statement above a problem? What is the burden of maternal and child morbidity and mortality in Asia in the context of HIV, Syphilis, Hepatitis C? What are the fall-outs from the supposed low uptake of available screening services? What do we stand to lose if this review is not done to better understand barriers and facilitators that can help inform useful interventions?

Response: We have clarified and expanded the introduction by highlighting the importance of this review (lines 88-92) and giving the reasons why a literature review in Asia is needed (lines 102-112). We now detail, in the introduction, the burden of maternal and infant morbidity and mortality in Asia within the limits of available data. We have also explained the consequences of the low uptake of screening services (lines 70-75).

Can the authors rewrite the Methods section without ‘We’?

Response: We decided to keep this section written in active voice as it is easier to understand and saves words. It is nowadays recommended for scientific writing in biomedical journals. 

Line 84. We used a very inclusive search strategy to ensure that no item was missed…..Can you give a 100% guarantee no item was missed?

Response: We removed this sentence from the manuscript.

There appears to be something in Line 92 that is not supposed to be there.

Response: We removed this from the manuscript.

Inclusion and Exclusion Criteria: The inclusion and exclusion criteria appear to be ‘all over the place’. They can be made more focused. We did not include studies investigating antenatal screening for other STDs……..This does not qualify as an exclusion criterion because you have earlier specified that you are dealing with HIV, Hep B and Syphilis.

Response: We agree and have rewritten the eligibility criteria for study inclusion using the acronym SPlDER: S sample; P phenomenon of interest; D design; E evaluation; R research type.

The data extraction sub-section has nothing on “facilitators”

Response: In the methods, we have added a subsection on the extraction of data and explained how we focus upon both barriers and facilitators to antenatal screening within the structure of Andersen’s framework (Table 3).

Line 121…..the authors may want to justify the choice of Andersen’s conceptual model over other models they could have used.

Response: Thank you for your suggestion. We chose the Andersen conceptual model because it provides an understanding of how individuals and environmental factors influence health behaviours. This theoretical framework is widely used in literature reviews on healthcare utilisation. This has been justified in lines 147-150.

For Table 1, I think the year the study was conducted ought to be in separate column by itself. This will enable readers to relate them to implementation of the MDGs and contextualize them. 

Responses: We agree with this suggestion and a separate column for the date has been added to Table 1.

I have some questions relating to Fig.1;

……you mentioned Google Scholar but I don’t see in the flow diagram

…….I am struggling to understand how you excluded 546 articles because the studies were not conducted in Asia when ‘Asia’ should have been a key part of your search strategy. Kindly elaborate on how this happened.

Responses: We have modified Figure 1 to reflect the new search strategy. With the new search strategy, only four articles were found to have been conducted outside Asia because the term "Asia" appeared in their abstracts.

Lines 210/211…….please give more meaning to ‘perceptions of poor healthcare support’ and ‘concerns about the reactions of healthcare workers

Response: We agree and have clarified the “perception of poor healthcare support” and the “concerns about the reactions of healthcare workers” (lines 276-291).

Secondly, of the 16 articles, 15 were on HIV and 1 was on Syphilis with nothing on Hepatitis B. In this context, I fail to see how you can make any reasonable pronouncements on Syphilis and Hepatitis B screening uptake. On this basis, I suggest you drop off Syphilis and Hep B and make your work entirely about the uptake of HIV screening than about STDs and appropriately reorient your discussion to that effect.

Response: We agree with your suggestion with respect to the discussion and conclusions and have rewritten these sections with respect to HIV only. However, the new search showed up three papers on syphilis and one on hepatitis B, so we believe it is important to highlight this gap and summarise the limited evidence in the results.

We hope that you will be satisfied with the amendments made. If there are any further issues do not hesitate to get in touch. We would like to thank you again for your time and consideration of our manuscript. 

Yours sincerely,

Lucie Sabin (on behalf of all co-authors)

---

## [Decision Letter · Decision Letter 1]

2 Jan 2024

PONE-D-23-03027R1A systematic review of barriers and facilitators to antenatal screening for HIV, syphilis or hepatitis B in Asia: perspectives of pregnant women, their relatives and health care providers.PLOS ONE

Dear Dr. Sabin, Thank you for resubmitting your manuscript to PLOS ONE. After careful consideration, we feel that it has merit but does not fully meet PLOS ONE’s publication criteria as it currently stands. The manuscript was sent for further review - an initial reviewer advised minor revisions (not accept) and a third (new) reviewer has indicated major revision. The original second reviewer was not available for re-review. You have the benefit then of three careful reviews. Therefore, we invite you to submit a revised version of the manuscript that addresses the points raised during the review process.

We look forward to receiving your revised manuscript.

Kind regards,

Steve

Stephen Michael Graham, FRACP, PhD

Academic Editor

PLOS ONE

Additional Editor Comments:

This submission was reviewed for a second time - and again a decision of Major Revision has been made. If you decide to resubmit, then there will need to be clear evidence that you have addressed all concerns of the reviewers for it to be considered for re-review - and then further review will be required anyway.

Reviewers' comments:

Reviewer's Responses to Questions

**Comments to the Author**

1. If the authors have adequately addressed your comments raised in a previous round of review and you feel that this manuscript is now acceptable for publication, you may indicate that here to bypass the “Comments to the Author” section, enter your conflict of interest statement in the “Confidential to Editor” section, and submit your "Accept" recommendation.

Reviewer #2: (No Response)

Reviewer #3: (No Response)

2. Is the manuscript technically sound, and do the data support the conclusions?

Reviewer #2: Partly

Reviewer #3: Yes

3. Has the statistical analysis been performed appropriately and rigorously? 

Reviewer #2: N/A

Reviewer #3: Yes

4. Have the authors made all data underlying the findings in their manuscript fully available?

Reviewer #2: Yes

Reviewer #3: No

5. Is the manuscript presented in an intelligible fashion and written in standard English?

Reviewer #2: Yes

Reviewer #3: Yes

6. Review Comments to the Author

Reviewer #2: Comments on manuscript PONE-D-23-03027R1

I am grateful to the authors for incorporating the previous comments made. The manuscript looks more refined now but will still need to be improved in some aspects as listed below.

After they have worked on these comments, I believe the work can be accepted for publication

Abstract

• Line 21………I am wondering if Antenatal screening alone is enough for PMTCT. Shouldn’t it be accompanied by treatment?

• Line 26…..read it again and rectify the grammatical error there….similar error in Line 30/31

• Line 27……..what you sought for in those published articles should be in this line.

• There is nothing about “barriers” in the Results section. If there is, it needs to be made more explicit

Introduction

• Line 73/74……can the authors show proof that there is a low uptake of STDs screening?

Methods

• Who or which persons conducted the initial search? (show with initials)

• Who is the third reviewer?

• Line 133……what is the use of the boldened Error statement there?

• In SPIDER, you defined SAMPLE to include women of childbearing age. I reckoned this work was about Antenatal Screening of Pregnant women. Please clarify the need to include women of childbearing age.

• Review the Data Extraction section for some grammatical omissions and errors. Line 150 is missing “of”. Line 151 …..’categorizes’ instead of ‘categories’.

• Line 160……’that’ instead of ‘who’

Discussion

Line 321…..rephrase to read “…….allow conclusions to be drawn effectively about HIV alone”

Line 340/341…….there is free screening of HIV, Hep B and Syphilis in many parts of sub-Saharan Africa already.

Line 348……On heterogeneity……there were 19 quantitative studies and you could have evaluated heterogeneity statistically to enable you make a more refined pronouncement on the subject

The Conclusion can be better written…..with emphasis on what specifically needs to be done and by which organization or department or health agency

Reviewer #3: GENERAL COMMENTS:

This review deals with an important topic that could be very useful in the prevention of HIV, syphilis and hepatitis B and transmission of these infections to infants. However, there are several major limitations to the quality of the review which hinder its relevance and applicability. The initial most striking feature of this review is that all four authors are affiliated with only one institute and this institute is in a high-income country that is not in Asia (used as general term here as the authors do not define Asia in their manuscript). Do any of these authors have lived experience of ANC, or healthy policy or practice in the region they are reviewing? If so, it would be helpful to have this information somewhere. Furthermore, the review only includes manuscripts published in English, a major limitation given the region being considered. there are excellent research Institutes throughout Asia, and no doubt this review would be enhanced if it included some collaborations within Asia to increase available grey literature and other studies that may not have been in their search methods.

DETAILED COMMENTS:

ABSTRACT:

“Despite improvements” is vague, some time reference of stats would be helpful here.

STIs is more commonly used now, rather than STDs.

“Antenatal screening” (Sabin, p. 1)

• and treatment. without treatment, screening won’t prevent transmission.

Methods paragraph. typo ‘conducted’ included twice in 1st and last sentence.

What is the definition of ‘Asia’? This should be included in the abstract.

Results section: please define in predisposing characteristics, enabling factors and need factors who you are referring to. The pregnant woman? The health worker?

INTRODUCTION:

“antenatal screening” (Sabin, p. 12) Line 62. And treatment, screening alone will not prevent transmission.

“infected women may transmit infections to their sexual partners or children” (Sabin, p. 13) Line 71. Please rephrase. Women are often infected by their partner, only saying women may give it to their partners overemphasizes their responsibility. What do you mean by infecting their children? Do you mean by MTCT? If so please be specific.

“Meanwhile, it encourages the participation of women living with HIV” (Sabin, p. 13) Line 86. prevention of MTCT of HIV, syphilis and hepatitis B is a shared responsibility, men and communities should also be encouraged to participate.

Preventing male transmission to women during sex, as well as preventing community transmission, of HIV, Syphilis and Hep B is also an effective method of preventing neonatal and infant infections. Whilst I recognise this is not the focus of the review, it should at least be mentioned to prevent misunderstandings and reduce stigma. Overly focusing on pregnant women being the source of transmission to their infants misses an opportunity to emphasise that they are not always the original source of the infection and may not have been able to negotiate appropriate protection for themselves in order to avoid infection.

“An estimated 10,000 new HIV infections occurred 56 among children aged 0–14 years in the Asia Pacific region in 2017” (Sabin, p. 12) Line 56/57. What is the number of infants infected with HIV due to MTCT? You mentioned 10,000 children infected between 0 and 14 years, but clearly not all of these are necessarily due to MTCT.

METHODS:

Line 122, word repetition.

Research type. Why were the articles limited to English? Given most countries in Asia have a primary language other than English this seems a big problem / barrier to identifying relevant research.

RESULTS:

Table 3. It would be helpful to also have a column of disease studied in this table.

Line 208. Whose knowledge are you referring to?

Paragraph re male partner’s opinion. In some countries mentioned it may be impossible for a woman to be screened without the express permission of the husband. It would be useful to contrast findings against legal framework for relevant countries as the approach to overcoming this barrier would be very different.

DISCUSSION:

Line 317. Given this review was limited to the English language I do not agree with it being referred to as a “comprehensive synthesis.”

Terminology used is not consistent regarding if this is a scoping review, narrative review or systematic review.

Paragraph 2. Part of the justification for this review was that findings in Asia may differ from that already published in sub-Saharan Africa. Given this, it would be interesting to understand the similarities and differences in more detail in this paragraph.

Line 330/331. Can you reference other differences in ANC screening or barriers that may support this statement?

Line 336/337. It is likely that training programs already exist, could you please highlight what efforts are already made in these settings before suggesting interventions. Again line 340/341 calls for free screening, this would be more helpful if information regarding whether this does or does not exist in the areas included in studies would be more meaningful.

The limitations paragraph needs to mention the limitation of including only English language and the apparent lack of inclusion of experts from the region.

CONCLUSION:

“and STDs” (Sabin, p. 30) Line 454. Please rephrase, you do not address all sexually transmitted infections.

“systematic review” (Sabin, p. 30) Line 354. Be consistent with use of terms narrative or systematic review.

In terms of translating these findings into practice it would be helpful, if possible, to comment in the conclusion as to which factors appeared to be the largest barriers. It may be that this varies in different countries, or at the sub district level. In addition to reviewing studies that look at implementing screening (and treatment), it would be more helpful to also know/contrast this with which countries have policies for ANC screening and treatment and if this is meant to be free or fee for service.

7. PLOS authors have the option to publish the peer review history of their article (what does this mean?). If published, this will include your full peer review and any attached files.

Reviewer #2: No

Reviewer #3: No

---

## [Author Response · Author response to Decision Letter 1]

21 Feb 2024

Dear Doctor Ampofo,

We thank you and the reviewer for your time and generous comments provided on our manuscript PLONE-D-23-03027 entitled "Barriers and facilitators to antenatal screening for sexually transmitted diseases in Asia: a scoping review", and we thank you for the opportunity to address these comments. After consideration of your comments, we have made several improvements. 

The comments provided are shown in bold below, with our responses in italics. 

Editor's Comments to the Author:

Response: We have ensured that the manuscript meets the stylistic requirements of PLOS ONE.

2. Please include a copy of Tables 4 and 5 which you refer to in your text on page 6.

Response: A copy of tables 4 and 5 has been included under the name S2 Table and S3 Table.

3. Please include captions for your Supporting Information files at the end of your manuscript, and update any in-text citations to match accordingly.

Response: Captions for our Supporting Information files have been included at the end of the manuscript.

Reviewers’ Comments to the Author:

Reviewer #1:

The topic is of great interest, however the manuscript seems hurriedly put together. Concepts involved such as Andersen's model are not clearly and coherently applied in the course of the write-up. 

Response: Thank you for your comment. We now define Andersen’s model (lines 148-153) and clearly and systematically apply the concepts cited. We have added Table 3 and systematically structured our results according to each component of Andersen’s model. The "facilitators" have been taken into account appropriately in the review.

The title for example does not capture the full essence of the review - it seems to be based on the "regional framework for the triple elimination of MTCT of HIV, Hep B and syphilis in Asia and Pacific" (lines 99 - 101)- that should have reflected clearly in the title. Such an understanding should have been provided briefly in the "Background" section of the script.

Secondly, the target population of the review was not clear right from the "title". Who did the review focus on - pregnant women, healthcare providers, relatives, etc. or all of them? That is not clear from the manuscript.

Response: We have changed the initial title from “Barriers and facilitators to antenatal screening for sexually transmitted diseases in Asia: a scoping review” to “A systematic review of barriers and facilitators to antenatal screening for HIV, syphilis or hepatitis B in Asia: perspectives of pregnant women, their relatives and health care providers.” The role of the WHO framework as the basis for this study has also been clarified in the background section (lines 76-89).

The objectives of the review as stated in lines 74 and 75 are not meant for a scoping review, but some other approaches that would generate clear evidence such as a review of qualitative evidence.

Response: On the basis of all the reviewers' comments, we decided to carry out a systematic analysis of the literature rather than a scoping review, using a second person to screen and review articles in duplicate and by searching other databases. We adapted the objectives to those of a systematic literature review (lines 113-115).

The methods section has lots of lapses.

a. A key concept such as the "population of the review" is not defined. The "search strategy" should have been systematically presented - e.g., keywords/ subject headings/ index terms that were used, an example of a search strategy of at least one of the databases should have been placed in the appendices; the search should have been as exhaustive as possible - searching two databases does not sound exhaustive enough.

Response: Thank you for this comment. To make the search as exhaustive as possible and in line with other reviews of this nature, we searched Ovid (MEDLINE, Embase, PsycINFO), Scopus, Global Index Medicus and Web of Science. Our database search strategy has been included in S1 File.

b. The "inclusion and exclusion criteria" should have been clearly defined each criterion - this subsection at best is confusing to the reader as it does not serve its purpose of clearly pointing out how studies were included in the review. Studies of experimental designs were excluded with no apparent reasons as to why they were.

Response: The eligibility criteria for the inclusion of studies have been rewritten and tabulated in Table 2 using the acronym SPlDER: S sample; P phenomenon of interest; D design; E evaluation; R research type. Our original inclusion/exclusion criteria were unchanged except that we also decided to include experimental design studies.

c. Under "Data extraction", the Andersen's framework was said to have been the guide - it would have been great to have this clearly "tabulated" with "findings" related to each component of the framework duly and systematically presented. The quality of studies was assessed, but this outcome did not feature further into any decision-making or discussion as to how study quality influenced the review process. 

Response: We agree that a more systematic application of Andersen’s framework improves the paper so have rewritten the results section to clearly reflect each of the elements of the Andersen framework and added Table 3. As the aim of the review was to describe and synthesise a body of literature and not to determine effect size, we did not exclude studies on the basis of their quality assessment (lines 160-161) but we have summarised the quality assessment in S2 and S3 Tables to enable readers to see the quality of the evidence included in the review. 

d. Andersen's model as defined in lines 123 - 124 does not seem to be in sync with the cited publication (11), i.e., predisposing, enabling, and needs factors. "Enabling factors" were not given the prominence required. 

Response: We agree with this comment and have now given more explanation of the ‘enabling factors’ of antenatal screening (lines 151-152).

Aside the mention of "facilitators" on line 147, this key component of the review was hardly addressed. Furthermore in the results section, the nature of the findings as were presented did not necessarily aptly fit the defined factors under the Andersen model. 

Response: As explained above we have now explicitly defined the "facilitators" in Table 1 and entirely rewritten the results section to follow the factors defined in Andersen’s model.

In its current form, this manuscript in my view is not fit for publication.

Response: Thank you for your detailed comments. We hope that the changes you have made will enable the study to be published.

Reviewer #2:

Title…..perhaps the title should be restricted to HIV, Syphilis and Hepatitis B rather than ‘sexually transmitted diseases’.

Response: Thank you. We have changed the title to “A systematic review of barriers and facilitators to antenatal screening for HIV, syphilis or hepatitis B in Asia: perspectives of pregnant women, their relatives and health care providers.”

Introduction: Despite being a scoping review, a little more detail will improve the Background and situate arguments in better context.

Response: We have expanded the introduction to better place the study in context. We have also changed from a scoping to a systematic review.

Lines 50/51…….the authors can provide recent data on the morbidity and mortality they refer to…..first globally and in the Asian context Quantify the prevalence of these STDs in Asia (at least present data from some Asian countries).

Response: Data on mother-to-child transmission of HIV, syphilis and hepatitis B were added to the introduction. The prevalence of the STDs considered was quantified in the introduction section.

Lines 58/61…..can the authors assign, at least, an estimate of how many children are born to these STDs? Can they quantify antenatal screening for STDs in Asia? Can they give us an idea of how low is ‘low’.

Response: We have added an estimate of the number of children born with these STDs (lines 54-59), as well as a quantification of antenatal screening for STDs in Asia (lines 63-68).

I think it would be useful to give a brief overview of this WHO regional framework and what different prescriptions it gave compared to whatever existed before its formation

Response: Thank you for this suggestion. We have added an overview of the WHO regional framework and its main prescriptions (lines 76-87).

Lines 68/70……The authors may want to give more meaning to the listed ‘barriers’ and how they relate to uptake of HIV screening services. It would be particularly interesting to see what the story is for “health system and health care provider issues”. In many jurisdictions in sub-Saharan Africa, HIV screening is part of the antenatal care package and is offered using the opt-out model.

Response: Thank you for this suggestion. The obstacles listed have been detailed for greater clarity.

There is no literature review summarising the research-based evidence on barriers and facilitators to antenatal screening for STDs in the Asian context. Why is the statement above a problem? What is the burden of maternal and child morbidity and mortality in Asia in the context of HIV, Syphilis, Hepatitis C? What are the fall-outs from the supposed low uptake of available screening services? What do we stand to lose if this review is not done to better understand barriers and facilitators that can help inform useful interventions?

Response: We have clarified and expanded the introduction by highlighting the importance of this review (lines 88-92) and giving the reasons why a literature review in Asia is needed (lines 102-112). We now detail, in the introduction, the burden of maternal and infant morbidity and mortality in Asia within the limits of available data. We have also explained the consequences of the low uptake of screening services (lines 70-75).

Can the authors rewrite the Methods section without ‘We’?

Response: We decided to keep this section written in active voice as it is easier to understand and saves words. It is nowadays recommended for scientific writing in biomedical journals. 

Line 84. We used a very inclusive search strategy to ensure that no item was missed…..Can you give a 100% guarantee no item was missed?

Response: We removed this sentence from the manuscript.

There appears to be something in Line 92 that is not supposed to be there.

Response: We removed this from the manuscript.

Inclusion and Exclusion Criteria: The inclusion and exclusion criteria appear to be ‘all over the place’. They can be made more focused. We did not include studies investigating antenatal screening for other STDs……..This does not qualify as an exclusion criterion because you have earlier specified that you are dealing with HIV, Hep B and Syphilis.

Response: We agree and have rewritten the eligibility criteria for study inclusion using the acronym SPlDER: S sample; P phenomenon of interest; D design; E evaluation; R research type.

The data extraction sub-section has nothing on “facilitators”

Response: In the methods, we have added a subsection on the extraction of data and explained how we focus upon both barriers and facilitators to antenatal screening within the structure of Andersen’s framework (Table 3).

Line 121…..the authors may want to justify the choice of Andersen’s conceptual model over other models they could have used.

Response: Thank you for your suggestion. We chose the Andersen conceptual model because it provides an understanding of how individuals and environmental factors influence health behaviours. This theoretical framework is widely used in literature reviews on healthcare utilisation. This has been justified in lines 147-150.

For Table 1, I think the year the study was conducted ought to be in separate column by itself. This will enable readers to relate them to implementation of the MDGs and contextualize them. 

Responses: We agree with this suggestion and a separate column for the date has been added to Table 1.

I have some questions relating to Fig.1;

……you mentioned Google Scholar but I don’t see in the flow diagram

…….I am struggling to understand how you excluded 546 articles because the studies were not conducted in Asia when ‘Asia’ should have been a key part of your search strategy. Kindly elaborate on how this happened.

Responses: We have modified Figure 1 to reflect the new search strategy. With the new search strategy, only four articles were found to have been conducted outside Asia because the term "Asia" appeared in their abstracts.

Lines 210/211…….please give more meaning to ‘perceptions of poor healthcare support’ and ‘concerns about the reactions of healthcare workers

Response: We agree and have clarified the “perception of poor healthcare support” and the “concerns about the reactions of healthcare workers” (lines 276-291).

Secondly, of the 16 articles, 15 were on HIV and 1 was on Syphilis with nothing on Hepatitis B. In this context, I fail to see how you can make any reasonable pronouncements on Syphilis and Hepatitis B screening uptake. On this basis, I suggest you drop off Syphilis and Hep B and make your work entirely about the uptake of HIV screening than about STDs and appropriately reorient your discussion to that effect.

Response: We agree with your suggestion with respect to the discussion and conclusions and have rewritten these sections with respect to HIV only. However, the new search showed up three papers on syphilis and one on hepatitis B, so we believe it is important to highlight this gap and summarise the limited evidence in the results.

We hope that you will be satisfied with the amendments made. If there are any further issues do not hesitate to get in touch. We would like to thank you again for your time and consideration of our manuscript. 

Yours sincerely,

Lucie Sabin (on behalf of all co-authors)

---

## [Editor Report · Decision Letter 2]

22 Feb 2024

PONE-D-23-03027R2A systematic review of barriers and facilitators to antenatal screening for HIV, syphilis or hepatitis B in Asia: perspectives of pregnant women, their relatives and health care providers.PLOS ONE Dear Dr. Sabin, Thank you for resubmitting your manuscript to PLOS ONE. After careful consideration, we feel that you have addressed comments and suggestions of previous reviewers. I request that you consider comments below about clarity on age ranges and representativeness of the populations studied - if possible.

Kind regards,

Steve

Stephen Michael Graham, FRACP, PhD

Academic Editor

PLOS ONE

Journal Requirements:

**Additional Editor Comments:**

Thanks for resubmitting and addressing the comments of the reviewers so comprehensively.

I support publication - however, would it be possible to improve the reporting of these populations by age groups that they represent? Is that something that could be added in a specific column in Table for each study: age range OR what proportion were adolescent pregnancy for example.

Adolescent pregnancy is still common in the region and a very vulnerable population with known poorer outcomes for mother and baby. V neglected population and as at risk for such infections as other pregnant women but perhaps even less likely to be screened? There is a data gap.

If this is not possible, it may still be worth a comment in discussion to highlight lack of data by age, especially in this vulnerable group. a suggested ref for this would be Sabet F, et al. The forgotten girls: .....Lancet. 2023;402:1580-1596.

- - - - -

---

## [Author Response · Author response to Decision Letter 2]

29 Feb 2024

Dear Dr Graham,

We thank you and the reviewer for your time and comments provided on our manuscript PLONE-D-23-03027 entitled " A systematic review of barriers and facilitators to antenatal screening for HIV, syphilis or hepatitis B in Asia: perspectives of pregnant women, their relatives and health care providers", and we thank you for the opportunity to address these comments. After consideration of your comments, we have made several improvements. 

The comments provided are shown in bold below, with our responses in italics. 

Journal Requirements:

Response: The list of references has been examined. It is complete and correct, and no changes were required.

Additional Editor Comments:

Thanks for resubmitting and addressing the comments of the reviewers so comprehensively.

I support publication - however, would it be possible to improve the reporting of these populations by age groups that they represent? Is that something that could be added in a specific column in Table for each study: age range OR what proportion were adolescent pregnancy for example.

Adolescent pregnancy is still common in the region and a very vulnerable population with known poorer outcomes for mother and baby. V neglected population and as at risk for such infections as other pregnant women but perhaps even less likely to be screened? There is a data gap.

If this is not possible, it may still be worth a comment in discussion to highlight lack of data by age, especially in this vulnerable group. a suggested ref for this would be Sabet F, et al. The forgotten girls: .....Lancet. 2023;402:1580-1596.

Response: Thank you for highlighting the importance of this research gap. Unfortunately, it was not possible to report the proportion of adolescent pregnancies in each study, as this information was not always included in the articles. However, we have added a paragraph in the discussion section on the lack of age-specific data, particularly for this vulnerable group of pregnant women, and the importance of filling this data gap (lines 388 to 394).

We hope that you will be satisfied with the amendments made. If there are any further issues do not hesitate to get in touch. We would like to thank you again for your time and consideration of our manuscript. 

Yours sincerely,

Lucie Sabin (on behalf of all co-authors)

---

## [Editor Report · Decision Letter 3]

1 Mar 2024

A systematic review of barriers and facilitators to antenatal screening for HIV, syphilis or hepatitis B in Asia: perspectives of pregnant women, their relatives and health care providers.

PONE-D-23-03027R3

Dear Dr Sabin,

We’re pleased to inform you that your manuscript has been judged scientifically suitable for publication and will be formally accepted for publication once it meets all outstanding technical requirements.

Kind regards,

Stephen Michael Graham, FRACP, PhD

Academic Editor

PLOS ONE
---

## [Editor Report · Acceptance letter]

10 May 2024

PONE-D-23-03027R3 

PLOS ONE

Dear Dr. Sabin, 

I'm pleased to inform you that your manuscript has been deemed suitable for publication in PLOS ONE. Congratulations! Your manuscript is now being handed over to our production team.

Kind regards, 

on behalf of

Dr. Stephen Michael Graham 

Academic Editor

PLOS ONE